# A Novel Frameshift *CHD4* Variant Leading to Sifrim-Hitz-Weiss Syndrome in a Proband with a Subclinical Familial t(17;19) and a Large dup(2)(q14.3q21.1)

**DOI:** 10.3390/biomedicines11010012

**Published:** 2022-12-21

**Authors:** Jorge Diogo Da Silva, Natália Oliva-Teles, Nataliya Tkachenko, Joana Fino, Mariana Marques, Ana Maria Fortuna, Dezso David

**Affiliations:** 1Centro de Genética Médica Jacinto de Magalhães (CGM), Centro Hospitalar Universitário do Porto (CHUPorto), 4099-001 Porto, Portugal; 2Life and Health Sciences Research Institute (ICVS), School of Medicine, University of Minho, 4710-057 Braga, Portugal; 3ICVS/3B’s—PT Government Associate Laboratory, 4806-909 Braga, Portugal; 4UMIB—Unidade Multidisciplinar de Investigação Biomédica, ICBAS—Instituto de Ciências Biomédicas Abel Salazar, Universidade do Porto, 4050-345 Porto, Portugal; 5ITR—Laboratory for Integrative and Translational Research in Population Health, 4050-600 Porto, Portugal; 6MEDCIDS—Departamento Medicina da Comunidade, Informação e Decisão em Saúde, Faculty of Medicine, University of Porto, 4200-450 Porto, Portugal; 7Department of Human Genetics, National Health Institute Doutor Ricardo Jorge, Av. Padre Cruz, 1600-609 Lisbon, Portugal

**Keywords:** Sifrim–Hitz–Weiss syndrome, *CHD4*-associated ND phenotype, frameshift *CHD4* variant, familial translocation, *GSG1L2*, dup(2)(q14.3q21.1)

## Abstract

The genetic complexity of neurodevelopmental disorders (NDD), combined with a heterogeneous clinical presentation, makes accurate assessment of their molecular bases and pathogenic mechanisms challenging. Our purpose is to reveal the pathogenic variant underlying a complex NDD through identification of the “full” spectrum of structural genomic and genetic variants. Therefore, clinical phenotyping and identification of variants by genome and exome sequencing, together with comprehensive assessment of these and affected candidate genes, were carried out. A maternally-inherited familial translocation [t(17;19)(p13.1;p13.3)mat] disrupting the GSG1 like 2 gene (*GSG1L2*), a 3.2 Mb dup(2)(q14.3q21.1) encompassing the autosomal dominant OMIM phenotype-associated *PROC* and *HS6ST1* gene, and a novel frameshift c.4442del, p.(Gly1481Valfs*21) variant within exon 30 of the Chromodomain helicase DNA binding protein 4 (*CHD4*) have been identified. Considering the pathogenic potential of each variant and the proband’s phenotype, we conclude that this case basically fits the Sifrim–Hitz–Weiss syndrome or *CHD4*-associated neurodevelopmental phenotype. Finally, our data highlight the need for identification of the “full” spectrum of structural genomic and genetic variants and of reverse comparative phenotyping, including unrelated patients with variants in same genes, for improved genomic healthcare of patients with NDD.

## 1. Introduction

Neurodevelopmental disorders (NDD) are often associated with cytogenomically visible or cryptic structural variants (SVs: translocations, inversions, and complex SVs) and copy number variants (CNVs: insertions, deletions, and duplications). The genetic complexity of NDD, combined with a heterogeneous clinical presentation, creates several challenges that range from accurate assessment of the molecular bases and pathogenic mechanisms to effective potential therapies.

A causal relationship between a de novo balanced chromosomal abnormality (BCA) and the associated clinical phenotype can be established in up to 40% of cases [1]. Indeed, we recently reported five de novo BCAs identified in patients exhibiting ND phenotypes, and in three of these, the BCA breakpoints disrupted genes causing autosomal dominant (AD) NDD, namely SKDEAS (Skraban–Deardorff syndrome) (OMIM #617616), *ANKS1B* haploinsufficiency syndrome, and KBG syndrome (OMIM #148050) [2]. Familial BCAs are rarely associated with clinical phenotypes. However, this may not be the case in families with cumulative, multilocus, or multigenic genomic variants, or inherited unbalanced derivative chromosomes of BCAs.

Furthermore, mainly due to incomplete annotation of the human genome and methodological difficulties, prediction of the pathogenic effect of even large novel CNVs or of distinct multilocus genomic variants is considerably difficult. Reverse phenotyping of probands and their relatives as well as comparison of their clinical features with those of unrelated patients with similar pathogenic variants within the referred genes can significantly improve this predictive analysis. A phenotypic features similarity search, based on information content, using the Human Phenotype Ontology standardized terminology and HPOSim tool, as implemented in SVInterpreter, provides a valuable computational approach for human disease phenotypes association predictions [3,4,5].

In addition to SVs and CNVs, assessment of the genetic etiology underlying NDD has to include the “full” spectrum of genetic variants, including in InDels (insertion and deletions of 1 to 50 bp in length) and single nucleotide variants (SNVs). Comprehensive genome sequencing (GS) approaches are able to identify the full spectrum of genomic variants, which is valuable towards improved genomic-based healthcare [2,5,6].

In this study, we report on a proband exhibiting a complex NDD associated with multilocus genomic and genetic variants, namely, a maternally inherited familial translocation [t(17;19)(p13.1;p13.3)mat], a 3.2 Mb dup(2)(q14.3q21.1), and a novel frameshift c.4442del, p.(Gly1481Valfs*21) variant within exon 30 of the *CHD4*. Finally, we consider the pathogenic potential of each variant and the clinical phenotype revealed to be Sifrim–Hitz–Weiss syndrome (SIHIWES) or *CHD4*-associated NDD (OMIM #617159), caused by the pathogenic frameshift alteration.

## 2. Materials and Methods

### 2.1. Karyotyping, Genomic DNA Extraction and Sanger Sequencing

Karyotyping and genomic DNA extraction from peripheral blood lymphocytes were carried out according to conventional protocols [7,8]. Amplification of control, translocation and deletion-specific junction fragments for family analysis and Sanger sequencing (SS) were performed as previously described [2]. Amplification primers and conditions are summarized in Appendix A.

### 2.2. Genome and Exome Sequencing

Quality control of DNA samples prior to long-insert genome sequencing (liGS) library preparation, sequencing and bioinformatics analysis was carried out as described previously [2]. Resulting genomic libraries were sequenced using Illumina sequencing by synthesis chemistry on HiSeq 2000 Sequencing Platforms (Illumina, San Diego, CA, USA) with multiplex paired-end 26 bp-cycle sequencing.

The exome sequencing (ES) library was prepared using the SureSelect Human All Exon V6 system (Agilent, Santa Clara, CA, USA), and followed by massively parallel sequencing (Illumina, San Diego, CA, USA).

### 2.3. Sequencing Data Analysis and Variant Interpretation

Analysis of liGS data was extensively described previously [2,6]. Briefly, the 26 bp paired-end sequence reads were mapped against the reference human genome version GRCh38/Hg38 using BWAv0.7.12 [9]. The resulting BAM file was processed by an in-house scriptimproperCLAS.py, (https://github.com/DGRC-PT/improperCLAS, accessed on 14 December 2022) to select improper pairs: read-pairs mapped in different chromosomes, those with an above average insert-size, or unexpected read orientations. These reads were then sorted by chromosomes and genomic coordinates, and clustered together by position using readPairCluster [6].

Concomitantly, the proper pairs were selected, and submitted to cn.MOPSv1.24 for identification of deletions and duplications based on depth-of-coverage (DoC) [10]. Lastly, the results from cluster and coverage analysis were merged and manually filtered. For graphical representation of CNVs, the selected proper pairs file was submitted to CNView [11] for coverage plot draw, and to the UCSC genome browser (http://genome.ucsc.edu, accessed on 14 December 2022), for large region alignment visualization and plotting. Structural and CNVs were interpreted using the comprehensive, clinically oriented SVInterpreter Web-application [5].

ES reads were aligned to the same reference genome and identified InDels and SNVs were annotated. Interpretation of known and novel likely pathogenic variants were performed according to standard protocols. Variants in genes associated with intellectual disability, with or without syndromic features, were especially assessed.

The phenotype association prediction was based on the phenotype similarity score (PhenSSc) functionality of the SVInterpreter web-based tool [3].

## 3. Case Report

### 3.1. Clinical Description

A 13-year-old female patient was referred for Medical Genetics consultation due to intellectual disability (HP:0001249) and behavioral disorders. She has non-consanguineous parents, a healthy young adult sister, and a brother with several neurological malformations that led to perinatal death. Family history reveals a fatal stroke in her father in his thirties, Alzheimer’s disease in her paternal grandmother, and strabismus and glaucoma in her maternal grandmother. There was a history of social risk due to familial disaggregation and low socioeconomical status.

The patient’s pregnancy and birth were unremarkable, and she was born at term with adequate weight and stature, but increased head circumference. Her stature was normal throughout growth, but she showed obesity (HP:0001513, weight above the 99th centile) since early infancy. She also maintained the postnatal macrocephaly (HP:0000256), reaching a 60.0 cm head circumference in adulthood (>99th centile, +3.76 SDS), which was similar to her mother’s (58.0 cm, >99th centile, +2.32 SDS). During her development, she showed generalized hypotonia (HP:0001290) and a delay in milestones since early infancy (Global developmental delay HP:0001263), being able to sit independently at 13 months, walk independently at 24 months and say her first words at 24 months of age (motor and speech delay HP:0001270 and HP:0000750, respectively). She was also diagnosed with a behavioral disorder (HP:0000708) in this period, which included severe bouts of auto- and hetero-aggressive behavior (HP:0000718), periods of increased agitation (HP:0000713), and a sleep disorder. She had a coarse facies (HP:0000280) but no relevant/specific dysmorphic features.

A cranial MRI at 14 months of age showed bilaterally widened cerebrospinal fluid spaces (CSF), with a predominance in the temporal area, but no parenchymal changes. She underwent additional diagnostic testing at 3 years of age, including an electroencephalogram and extended metabolic study (quantification of amino acids, organic acids, creatine, guanidinoacetic acid, and carbohydrate-deficient transferrin), which were within normal parameters.

At 8.5 years old, she was submitted to extensive ophthalmologic and cardiologic assessments, including electrocardiography and echocardiography with dynamic studies, both of which were unremarkable. Her menarche occurred at 13 years, with adequate puberty onset and development. During adolescence her behavioral issues worsened, namely her bouts of aggressive behavior and agitation, manifesting a very low threshold of frustration. She was additionally diagnosed with generalized anxiety disorder (HP:0000739) and severe emotional lability (HP:0000712) that led to dysregulation and instability of relationships with her family and friends. There was no record of behavioral disinhibition, depressive or psychotic disorders, nor of suicidal ideation. Several neurotropic drugs were tried during this period, such as anxiolytics, antidepressants and antipsychotics, with partial effectiveness in regulating the behavioral abnormalities. She underwent two assessments of cognitive abilities using the Wechsler Intelligence Scale for Children, 3rd edition (WISC-III) at 13 and 15 years old: both evaluations showed a significantly decreased general, verbal and performance intelligence quotient (63, 70, and 63, respectively), establishing the diagnosis of intellectual disability (ID HP:0001249). She was last examined in our consultation at 16-years-old. Presently, as a young adult, she maintains her psychiatric comorbidities for which she is currently taking risperidone but is able to maintain a functional and somewhat independent life.

### 3.2. Identification of SV and CNV Breakpoints at Nucleotide Resolution

G-banding karyotype of the proband revealed an apparently balanced chromosomal translocation between the short arms of chromosomes 17 and 19 [46,XX,t(17;19)(p13;p13.3)]. This abnormality was inherited from her mother, who had no relevant medical history.

Mapping and identification of the breakpoints at nucleotide resolution by liGS, followed by SS of the breakpoint spanning amplicons, defines the 17p13.1 breakpoint at NC_000017.11:g.9,814,480, within IVS1 of GSG1 like 2 gene (*GSG1L2*), whereas the 19p13.3 breakpoint at NC_000019.10:g.6,570,027 is within a low complexity region comprising several LINE elements. Deletions were identified at both breakpoint junctions: a 5.3 kb (NC_000017.11:g.9,814,480_9,819,771del) at 17p13.1 and a 3.2 kb (NC_000019.10:g.6,570,027_6,573,218del) at 19p13.3 (Figure 1). Hence, the translocation was classified as unbalanced and revised as seq[GRCh38] t(17;19)(19pter → 19p13.3::17p13.1 → 17qter;17pter → p13.1::19p13.3 → 19qter)mat.

Furthermore, an outward-facing read-pairs cluster at chromosome 2 q14.3q21.1 indicated a duplication of over 3 Mb in size. The duplication was also identified by DoC and confirmed by SS as 3,195,005 bp in size (NC_000002.12:g.125,920,700_129,115,703dup) (Figure 2). The upstream and downstream duplication breakpoints disrupt AluY repetitive elements in the same orientation (class SINE, family Alu) that spans 317 (chr2:129,115,639_129,115,935) and 297 bp (chr2:125,920,637_125,920,953), respectively. Hence, the mechanism of this rearrangement is non-allelic homologous recombination between these two AluY elements [12].

Segregation analysis, based on the translocation and deletion-specific junction fragments, confirmed that the mother is a carrier of the t(17;19) but not of the dup(2)(q14.3q21.1) (Appendix A); therefore, the latter is either of paternal origin or de novo.

### 3.3. Identification of Additional Genetic Defects

As the pathogenic effect of the identified genomic variants was unclear and no additional, potentially pathogenic CNV could be identified, ES was performed, with an average coverage of 125×. This analysis revealed a novel variant in exon 30 of the Chromodomain helicase DNA binding protein 4 (*CHD4*, OMIM *603277, NM_001273.5), NC_000002.12:g.6,582,210del; c.4442delG, p.(Gly1481Valfs*21). Afterwards, as an independent assay replacing Sanger sequencing, the frameshift variant was confirmed in the liGS data; however, due to low sequence coverage of liGS, this frameshift single-nucleotide deletion was identified only in a single GS read (Figure 3). This variant is within a domain of unknown function (DUF) 1086 in the C-terminal region of *CHD4* and is predicted to result in nonsense-mediated decay [13].

### 3.4. Characterization of the Genomic and Genetic Variants and Affected Candidate Genes

The 17p13.1 breakpoint spanning *GSG1L2*, which predictably encodes an integral component of the plasma membrane, is tolerant to LoF variants [o/e = 0.65 (90% CI 0.37–1.22); pLI = 0.00]. No known disease-related biological function has been ascribed to this gene.

Subsequently, genes localized within the breakpoint spanning topologically-associated domains (brTADs) in human embryonic stem cells (hESCs) [14], were also evaluated. Although within the 17p13.1 hESCs brTAD (NC_000017.11:g.9,800,000_10,640,000) three myosin heavy chain genes associated to AD disorders have been localized, none of these shown significant phenotypic similarity to the proband’s phenotype (Appendix A). From the 19p13.3 hESCs brTAD (NC_000019.10:g.5,760,000_7,040,000), two genes associated to AD disorders have been localized. Tubulin beta 4A class IVa (*TUBB4A*, OMIM *602662), which is a member of the beta tubulin family, a subunit of microtubules, is reported to cause AD Dystonia 4, torsion (OMIM #128101) and AD Leukodystrophy, hypomyelinating, 6 (OMIM #612438) [15]. The latter shows a significant PhenSSc of 1.05 (*p* = 0.00733; MaxSSc 3.02; MaxDiseaseSSc 2.89) with the proband’s phenotype (Appendix A). The second gene in question is Complement *C3* (OMIM *120700).

Within the 3195 kb duplicated region [dup(2)(q14.3q21.1)], six genes associated with OMIM phenotypes have been identified, namely *GYPC* (OMIM *110750), *BIN1* (OMIM *601248), *ERCC3* (OMIM *133510), *PROC* (OMIM *612283), *LIMS2* (OMIM *607908), and *HS6ST1* (OMIM *604846) (Figure 2 and Appendix A). Heterozygous pathogenic variants within *PROC* and *HS6ST1* have been reported to cause AD thrombophilia due to *PROC* deficiency (OMIM #176860) and Hypogonadotropic hypogonadism 15 with or without anosmia (OMIM #614880), respectively. Predictably, none of these genes has pathogenic implication in the clinical phenotype currently presented by the proband. No similar duplications were reported in either ClinVar and DECIPHER databases; however, the 3.5 Mb duplication reported at the ClinVar database, [Accession VCV000149324.2, (chr2:127,063,206-130,527,454)x3) with unknown significance, partially overlaps the one reported in our patient.

The heterozygous frameshift variant c.4442delG, p.(Gly1481Valfs*21) in the *CHD4* has not been previously reported. This variant fulfilled the following ACMG SNV criteria [16]: PVS1 (null variant in a gene where loss-of-function is a known mechanism of disease), PM2 (extremely low frequency, as it was not detected in population databases), PM4 (protein length changes as a result of in-frame deletions/insertions in a non-repeat region or stop-loss variants) and PM6 (assumed de novo, but without confirmation of paternity and maternity). Hence, this variant is classified as pathogenic.

This gene encodes the chromatin remodeler protein *CHD4*, which is a core component of the nucleosome remodeling and histone deacetylase repressor complex, involved in epigenetic regulation of gene transcription [17,18]. *CHD4* is sensitive to LoF variants o/e = 0.09 (90% CI 0.06–0.16) and is associated with clinically highly variable, multisystemic Sifrim–Hitz–Weiss syndrome (SIHIWES; Appendix A), also known as *CHD4*-NDD [17,19].

The proband’s phenotype similarity score, calculated against the SIHIWES clinical synopsis at OMIM #617159, was low and without statistical significance [PhenSSc 1.29 (*p* = 0.1960; MaxSSc 3.02; MaxDiseaseSSc 3.64)]. Therefore, the clinical features reported in 44 patients with SIHIWES or *CHD4*-NDD were extensively revised and subdivided according to the pathogenic *CHD4* variant (Appendix A) [19,20,21,22,23,24]. Indeed, the PhenSSC calculated against the revised clinical features reference list was 2.33, and near to statistical significance (*p* = 0.0587; MaxSSc 3.02; MaxDiseaseSSc 4.2) (Table 1 and Appendix A).

Therefore, this LoF *CHD4* variant is the most likely cause of the patient’s phenotype, establishing the diagnosis of SIHIWES or *CHD4*-NDD.

## 4. Discussion

Exhaustive clinical phenotyping of a proband with a complex NDD, and identification of structural genomic and genetic variants by genome and exome sequencing, followed by comprehensive assessment of these and affected candidate genes, were performed.

Concerning the familial translocation, the disrupted 17p13.1 breakpoint spanning gene (*GSG1L2*) is tolerant to LoF variants, no gene causing AD developmental or NDD has been identified within the disrupted bpTADs, and the proband’s mother and maternal grandmother are healthy carriers of this translocation. Therefore, the t(17;19)(p13;p13.3) is most likely subclinical or nonpathogenic.

Regarding the 3195 kb dup(2)(q14.3q21.1), while the pathogenic effect of *PROC* deficiency is well known, the effect of increased *PROC* level, which circulates in blood as an inactive zymogen, is unknown [25]. To our knowledge, the reported pathogenic ClinVar duplication nssv13639329, encompassing the entire *PROC*, is over 241 Mb in size, whereas the DECIPHER CNVs 254,507 and 304,025, of 47.2 and 16.6 Mb, respectively, are currently reported with unknown pathogenicity. Similarly, the effect of the duplication on the significantly LoF intolerant *HS6ST1* is unknown [26]. A somehow comparable 119.8 kb duplication, encompassing the entire gene, was reported in a subject with neurological phenotype (nssv3451650). Therefore, we consider this duplication as a variant of unknown significance.

SIHIWES or *CHD4*-NDD is characterized by a highly variable clinical phenotype, but almost all patients have some degree of developmental delay and/or intellectual disability (Table 1) [18]. Five of the proband’s clinical features are among the most frequently reported features in SIHIWES patients (from 32% to 72%; Table 1), while only three of her clinical features (agitation, aggressive behavior, and emotional lability) are yet unreported in other patients. This is likely attributed to the syndrome’s phenotypic variability, to the complex neurological phenotype presented by the proband, but the modulating effect of the large duplication cannot be excluded. Other common phenotypic features related to this syndrome are congenital heart defects (68.2%), ophthalmological abnormalities (35.9%), and hearing impairment (26.8%). In addition, there is an increased risk for moyamoya disease (8.6%) and of a progressive cerebrovascular disorder that mainly affects cervical arteries (20.5%) (Appendix A) [18,19,27].

According to the proband’s phenotype and PhenSSc, we conclude that this case fits the SIHIWES or *CHD4*-associated ND phenotype and reflects its clinical variability. Very small numbers of patients have been reported with truncating variants, making it difficult to assess phenotypic differences between patients carrying missense and truncating variants (Table 1 and Appendix A).

Regarding follow-up, the proband should be monitored for increased *PROC* activity and antigen level, and evaluated by a psychiatrist for management of behavioral issues, as well as by a neurologist due to the risk of moyamoya disease.

In conclusion, we excluded the pathogenic effect of a familial translocation and identified a large duplication encompassing *PROC* and *HS6ST1* associated with AD OMIM phenotypes and a novel frameshift variant within exon 30 of *CHD4* as the most likely genetic basis of a complex NDD defined as SIHIWES. Finally, our data highlight the need for identification of the “full” spectrum of genomic/genetic variants and of reverse comparative phenotyping, including unrelated probands with variants in similar genes, for improved genomic healthcare of patients with NDD.

## Figures and Tables

**Figure 1 biomedicines-11-00012-f001:**
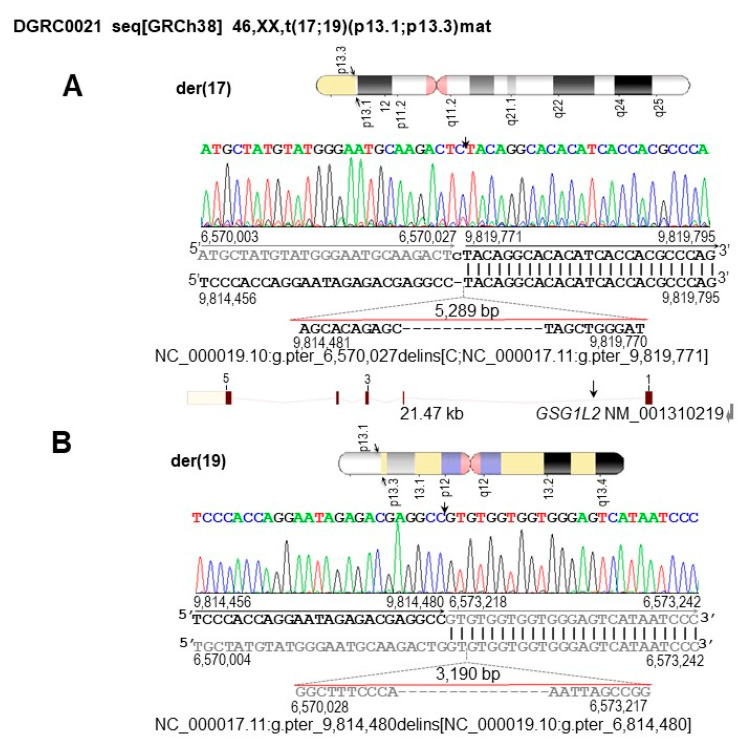
Characterization of the 46,XX,t(17;19)(p13.1;p13.3)mat breakpoint regions and affected genomic elements at nucleotide resolution. Arrows indicate breakpoint positions. The chromosome 17 sequence is in black whereas the chromosome 19 sequence is in gray. Vertical lines indicate identical nucleotides between derivative and reference chromosomes. Orientation of genomic sequences within the derivative chromosomes are indicated by color-coded arrows. (**A**) Ideogram of the der(17) chromosome and corresponding breakpoint region 17p13.1. The C nucleotide insertion is in lowercase. Below the sequence alignment, the 5289 bp deletion is depicted. The deletion is evidenced by the difference between the genomic position of the 17p13.1 breakpoints on the derivative chromosomes (g.9,814,480_9,819,771del). Below, schematic map of *GSG1L2*. Exons are depicted by numbers, whereas the position of the 17p13.1 breakpoint within IVS1 is indicated by an arrow. (**B**) Ideogram of the der(19) chromosome and corresponding breakpoint region at 19p13.3. Below the sequence alignment, the 3190 bp deletion is depicted. The deletion is evidenced by the difference between the genomic position of the 19p13.3 breakpoints on the derivative chromosomes (g.6,570,027_6,573,218del). Detailed ISCN- and HGVS-based descriptions of the der(17) and der(19) at nucleotide level are: NC_000019.10:g.pter_6,570,027delins[C;NC_000017.11:g.pter_9,819,771] and NC_000017.11:g.pter_9,814,480delins[NC_000019.10:g.pter_6,814,480]. The translocation is revised to seq[GRCh38] t(17;19)(19pter → 19p13.3::17p13.1 → 17qter;17pter → p13.1::19p13.3 → 19qter)mat.

**Figure 2 biomedicines-11-00012-f002:**
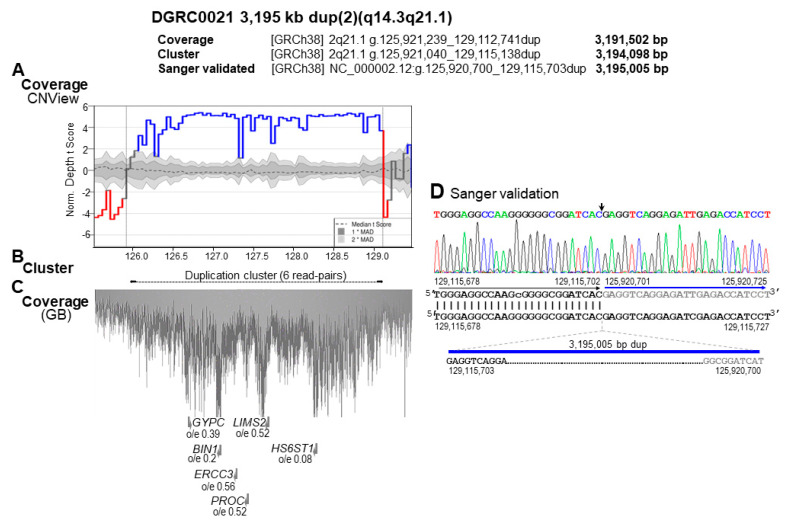
Illustration of the proband-specific dup(2)(2q14.3q21.1) of 3195 kb by genome coverage, read-pair cluster and SS. The size of the duplication varies according to the detection method and is indicated above. (**A**) Genomic coverage plot generated by CNView showing the 3,191,502 bp duplication. The horizontal black dashed line with darker and lighter gray shading indicates median coverage and deviation, respectively. Regions with a statistically significant increase in sequence coverage (α = 0.05, Bonferroni correction) indicating duplications are depicted in blue. (**B**) Arrows joined by a dashed line indicate the location and orientation of the read−pairs cluster identifying the 3194 kb dup, with respect to the human physical genome map and genome coverage of this region. (**C**) A picture from the genome browser illustrating the coverage of this region by sequencing read−pairs. Each dot represents genomic location of a sequence read. Below, folded gray arrows indicate the position of OMIM genes in sense and antisense orientations. Their LoF intolerance score, expressed as the o/e ratio of LoF variants, is stated below each gene. (**D**) Nucleotide sequence of the 3,195,004 bp dup(2)(q14.3q21.1) junction fragment aligned against the GRCh38 reference human genome. The downstream and upstream sequences at the duplication breakpoint are in black and gray, respectively. Vertical lines indicate identical nucleotides between the aligned sequences.

**Figure 3 biomedicines-11-00012-f003:**
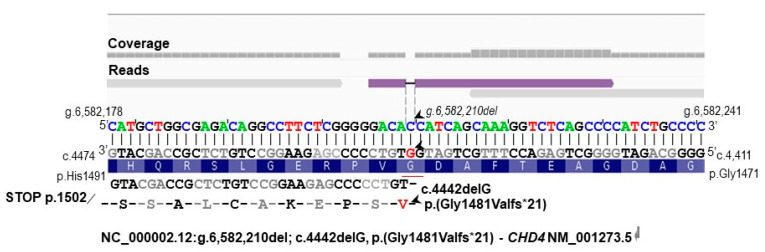
Alignment of liGS reads using the IGV tool, showing the frameshift deletion at NC_000002.12:g.6,582,210 in exon 30 of *CHD4*. The alignment of GS reads to the reference genome and the corresponding amino acids are indicated. The *CHD4* cDNA is in antisense orientation. The deleted G nucleotide is underlined in red. Below, the position of the c.4442delG relative to the cDNA (RefSeq NM_001273.5) and amino acid sequences is indicated. Arrowheads indicate the position of the frameshift deletion relative to the reference genome sequence, to the reference cDNA, and to the variant amino acid. Due to low sequence coverage, this is only of confirmatory value for the ES data.

**Table 1 biomedicines-11-00012-t001:** Comparison of the proband’s (DGRC0021) clinical features with those reported for SIHIWES or *CHD4*-NDD patients, subdivided according to the pathogenic variant type and the corresponding phenotype similarity score.

Proband’s Clinical Features, HPO Term	SIHIWES or *CHD4*-NDDpatients				
	OMIM #617159	All Variants (%)	Missense Variants (%)	Truncating Variants (%)	Splicing Variants
**Growth**					
Obesity, HP:0001513	No	3/36 (8.3)	3/31 (9.7)	0/3 (0.0)	0/2
**Head and neck**					
Coarse facies, HP:0000280	Yes	0/44 (0.0)	0/38 (0.0)	0/4 (0.0)	0/2
**Muscoloskeletal**					
Hypotonia, HP:0001252	Yes	18/35 (51.4)	16/30 (53.3)	2/3 (66.7)	0/2
Macrocephaly, HP:0000256	Yes	13/40 (32.5)	11/35 (31.4)	2/3 (66.7)	0/2
**Nervous system**					
Agitation, HP:0000713	No	0/44 (0.0)	0/38 (0.0)	0/4 (0.0)	0/2
Aggressive behavior, HP:0000718	No	0/44 (0.0)	0/38 (0.0)	0/4 (0.0)	0/2
Anxiety. HP:0000739	No	2/36 (5.6)	2/31 (6.5)	0/4 (0.0)	0/2
Emotional lability, HP:0000712	No	0/44 (0.0)	0/38 (0.0)	0/4 (0.0)	0/2
Global developmental delay, HP:0001263	Yes	2/44 (4.6)	2/38 (5.3)	0/4 (0.0)	0/2
Headache, HP:0002315	No	2/44 (4.5)	2/38 (5.3)	0/4 (0.0)	0/2
Hydrocephalus, HP:0000238	No	5/44 (11.4)	5/38 (13.2)	0/4 (0.0)	0/2
Intellectual disability, HP:0001249	Yes	20/34 (58.8)	19/31 (61.3)	1/1 (100)	0/2
Motor delay, HP:0001270	No	29/43 (67.4)	27/38 (71.1)	2/3 (66.7)	0/2
Speech delay, HP:0000750	No	31/43 (72.1)	28/38 (73.8)	3/3 (100)	0/2
**Phenotype similarity score ^a^ ** **PhenSSc (*p*-value)**	1.29 (0.20)	2.33 (0.06)	1.21 (0.07)	1.22 (0.01)	0.81 (0.16)

The ratio between variants-patients with a specific clinical feature per total number of patients-variants (n = 44) with available data is reported [19,20,21,22,23,24]. Additional information is available in Appendix A. ^a^ The PhenSSc and p-value obtained from the comparison of the proband’s features with the ones from SIHIWES syndrome clinical synopsis at OMIM #617159, and four categories of pathogenic variants.

## Data Availability

NCBI-GenBank accession numbers of the der(17), der(19) and dup(2)(q14.3q21.1)-specific junction fragment sequences are OP800824, OP800825 and OP800826, respectively.

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
