# Peer review of "A Novel Frameshift CHD4 Variant Leading to Sifrim-Hitz-Weiss Syndrome in a Proband with a Subclinical Familial t(17;19) and a Large dup(2)(q14.3q21.1)"

_biomedicines, 2022, doi:10.3390/biomedicines11010012_

Round 1

Reviewer 1 Report

The study by Jorge Diogo Da Silva et al. aimed to reveal a pathogenic variant underlying a complex neurodevelopmental disorder in a young adult female patient. This was done by identifying a 'full' spectrum of structural genomic and genetic variants. I find this case report very important and interesting because the molecular basis and pathogenetic mechanisms of neurodevelopmental disorders are challenging. The number of analyzed molecular targets is impressive. The authors identified a large duplication involving PROC and a novel frameshift variant in CHD4 exon 30 as the most likely genetic basis for the complex neurodevelopmental disorder defined as SIHIWES. Methods and results are clearly presented. I have no major complaints about this study. The only thing that needs to be completed is the age of the patient at the time of the examination.

Congratulations for your work.

Author Response

The study by Jorge Diogo Da Silva et al. aimed to reveal a pathogenic variant underlying a complex neurodevelopmental disorder in a young adult female patient. This was done by identifying a 'full' spectrum of structural genomic and genetic variants. I find this case report very important and interesting because the molecular basis and pathogenetic mechanisms of neurodevelopmental disorders are challenging. The number of analyzed molecular targets is impressive. The authors identified a large duplication involving PROC and a novel frameshift variant in CHD4 exon 30 as the most likely genetic basis for the complex neurodevelopmental disorder defined as SIHIWES. Methods and results are clearly presented. I have no major complaints about this study. The only thing that needs to be completed is the age of the patient at the time of the examination.

We appreciate and are thankful for the comments of the reviewer. We have now added the age of the patient at the referral (13 years of age) and at the time of the last examination (16 years of age) (Section 3.1, lines 125 and 167).

Reviewer 2 Report

Da Silva and colleagues describe a patient with a familial translocation, a large duplication on chromosome 2, and a pathogenic CHD4 variant. This is an interesting case, as the genomic variants would likely be at least variants of uncertain significance, and not all patients with these variants would have undergone exome sequencing. The authors provide a nice interpretation of their results to show that they believe only the CHD4 variant is causative.

I do have a few questions and some small edits:

1) Are there any clinical features that don’t fit CHD4-NDD and could fit with the other genomic changes? Or is this patient more severe than other reported patients (potentially the other variants are contributing?).

2) The CHD4 variant being in only one read is concerning, did you Sanger confirm? Hard to believe a variant with only one read! Also, could it be mosaic? While the phenotypic correlations are certainly compelling, NDDs are heterogenous (like you said in the manuscript!) so it's possible to see a correlation even with the wrong NDD.

3) Did you see if there were any similar duplications in ClinVar? If they were classified as benign that would support your story.

Overall, there are a few small formatting errors and use of phrases (listed below) that are distracting to the reader. A readthrough outloud (or using Microsoft Word's reading tool) would be useful.

I think with answering these questions this would be a nice manuscript. It is definitely helpful to contribute to the CHD4-NDD literature, and also supports genome sequencing, or at least sequential testing with exome sequencing.

Edits:

NDD (neurodevelopmental disorders) is typically used over ND or ND disabilities

Line 28 should read PROC gene

Line 44: I wouldn’t cite a poster for this, and I would list what the most common clinical feature is, as NDDs are typically a large proportion of clinical genetic testing. Or, you can remove this sentence entirely, as it does not change your story.

Line 52: spell out SKDEAS and KBGS. Put OMIM # in parentheses.

Line 53: this is a long sentence that reads a little confusing. I would change it to something like “On the contrary, familial BCAs are rarely associated with clinical phenotypes. However, this may not be the case in families with multiple genomic variants or inherited unbalanced translocation variants of BCAs.”

Line 67: again, a long sentence. Maybe change to “Comprehensive genome sequencing (GS) approaches are able to identify the full spectrum of genomic variants, which is valuable towards improved genomic-based healthcare.”  The following sentence is not necessary, of course GS needs GS data analysis/interpt! If you were showing us a novel tool it would make sense to include, but I don’t think its necessary.

Line 74: use 3.2 Mb instead of 3,195 kb

Line 77: include OMIM # (617159)

Line 81: “conventional protocols” is not sufficient, add a citation.

Line 88: saying “essentially” sounds like you’re leaving something out.

Line 89: is there a github/citation for improperCLAS.py? Otherwise we don’t really need the name of your in house pipeline.

Line 113: remove comma

Case Report: the HPO #s have different formatting (some w/ : some not, some with spaces some not). Be consistent.

Line 133: “In parallel” doesn’t make sense here. In parallel to what?

Line 140: “She performed” doesn’t make sense, I would change to “She underwent”. Same with line 144

Line 164: you say that the mother has “no relevant medical history,” but she is macrocephalic. Is that not relevant, since the proband is too?

Line 168: you don’t need to cite GeneCards

Line 169: should read “…is within a low complexity region…”

Line 170: use kb not bp

Figure 2: It would be good to include the genes encompassed by the duplication in the diagram.

Line 224: put # with OMIM # throughout manuscript

Line 228: https://nmdprediction.shinyapps.io/nmdescpredictor/ suggests that this variant is subjected to nonsense mediated decay, not truncation. Also not sure why you cited the Weiss paper there? 

Figure 3 is not super clear, maybe align the sequences differently?

Line 238: italicize GSG1L2. I would also change the sentence to say “GSG1L2, located at the 17p13.1 breakpoint, is predicted to be tolerant to loss-of-function…” Include the pLI score.

Genes should be in italics throughout manuscript.

Line 258: spell out AD

Line 263: change to “This variant fulfilled the following ACMG SNV criteria… and is considered pathogenic”

Line 274: CHD4-ND should be CHD4-NDD.

Line 298: use Mb not bp

Line 304: form should be from

Line 305: but other behavioral features are reported, correct? Perhaps these just haven’t been highlighted in the literature?

Table 1: cite where the 44 patients are from. You also might want to compare course facies to just dysmorphic features (abnormal facial shape: HP:0001999).

I wonder if you could combine Tables 1 and S1? The phenotypic match information is very useful!

Author Response

Da Silva and colleagues describe a patient with a familial translocation, a large duplication on chromosome 2, and a pathogenic CHD4 variant. This is an interesting case, as the genomic variants would likely be at least variants of uncertain significance, and not all patients with these variants would have undergone exome sequencing. The authors provide a nice interpretation of their results to show that they believe only the CHD4 variant is causative.

I do have a few questions and some small edits:

We appreciate and are thankful for the comments and thorough review. We have addressed the next suggestions and have changed the manuscript accordingly.

1) Are there any clinical features that don’t fit CHD4-NDD and could fit with the other genomic changes? Or is this patient more severe than other reported patients (potentially the other variants are contributing?).

We understand the reviewer’s concern, as the proband has psychiatric features that were not reported in other patients in the literature (“agitation (HP:0000713)”, “aggressive behavior (HP:0000718)” and “emotional lability (HP:0000712)”, as in Table 1). Those features are likely the result of the high variability of phenotype associated to CHD4-NDD, as 53 of 127 clinical features associated to this disorder (41.73%) were only detected in one patient (Table S4). Nevertheless, we cannot exclude the possibility of i) a modeling effect of the large duplication (Ouidja et al., 2022, Ref. 26); ii) the differences in the phenotypic description being derived from different interpretations by the clinicians; and iii) an environmental risk factor for these behavioral alterations , namely a social risk due to familial disaggregation and low socioeconomical status. Thus, a sentence in lines 130-131 was added:

There was a history of social risk due to familial disaggregation and low socioeconomical status.

Due to the high variability of the reported CHD4-NDD phenotypes, clinical severity evaluation becomes difficult. Still, the main measure that allows us to do so, is the IQ. In the WISC-III assessments at 13 and 15 years-old, our patient obtained an IQ of 63, which corresponds to “mild intellectual disability”. In fact, the patients in the literature mostly have “mild” or “moderate” intellectual disability, which is in line with what is seen with our patient. Finally, our patient achieved a more-or-less independent adult life (information added to Section 3.1 in line 169), which is something also observed in other CHD4-NDD patients in the literature.

Therefore, we concluded that the CHD4-NDD can basically explain the proband’s phenotype and the severity of the clinical case can reasonably be considered similar to the other reported cases.

2) The CHD4 variant being in only one read is concerning, did you Sanger confirm? Hard to believe a variant with only one read! Also, could it be mosaic? While the phenotypic correlations are certainly compelling, NDDs are heterogenous (like you said in the manuscript!) so it's possible to see a correlation even with the wrong NDD.

We appreciate the comment of the reviewer. Chronologically, we first performed long-insert genome sequencing (liGS) in the patient, with the main goal of characterizing the breakpoints of the apparently balanced translocation. Once the pathogenicity of the translocation and of the 3.2 Mb chromosome 2 duplication were excluded, we performed exome sequencing, with an average coverage of 125x, which identified the c.4442delG variant in CHD4. Afterwards, as an independent assay replacing Sanger sequencing, the c.4442delG variant was confirmed in the liGS data (Figure 3). We have changed the text in section 3.3. (lines 232-239) to clarify this, in accordance with the reviewer suggestion. The section now reads:

As the pathogenic effect of the identified genomic variants was unclear and no additional, potentially pathogenic CNV could be identified, ES was performed, with an average coverage of 125x. This analysis revealed a novel variant in exon 30 of the Chromodomain helicase DNA binding protein 4 (CHD4, OMIM *603277, NM_001273.5), NC_000002.12:g.6,582,210del; c.4442delG, p.(Gly1481Valfs*21).Afterwards. as an independent assay replacing Sanger sequencing, the frameshift variant was confirmed in the liGS data, however due to low sequence coverage of liGS, this was identified only in a single GS read (Figure 3).

3) Did you see if there were any similar duplications in ClinVar? If they were classified as benign that would support your story.

There are no similar duplications in the ClinVar database, with the closest one being a 3.5 Mb duplication which is distal to the one reported in our patient and affects a different set of genes (Accession VCV000149324.2, GRCh38/hg38 2q14.3-21.1(chr2:127063206-130527454)x3, unknown significance).

We also consulted the DECIPHER database, and found no duplication of a similar size. Smaller duplications within the duplicated region in our patient are either categorized as of unknown significance (for example, cases 451789 and 479870) or likely benign/benign (for example, cases 289947, 289557 and 487515).

We added this summarized information to Section 3.4 (lines 276-277): “No similar duplications were reported in both ClinVar and DECIPHER databases.

According to all the data available, we consider this duplication as a variant of unknown significance. To reinforce this idea we added (lines 316-320):

Similarly, the effect of the duplication on the significantly LoF intolerant HS6ST1, is unknown [26]. A somehow comparable 119.8 kb duplication, encompassing the entire gene, was reported in a subject with neurological phenotype (nssv3451650). Therefore, we consider this duplication as a variant of unknown significance.

Overall, there are a few small formatting errors and use of phrases (listed below) that are distracting to the reader. A readthrough outloud (or using Microsoft Word's reading tool) would be useful.

We have reviewed the language style and grammar as suggested, including the changes proposed in the next comments.

I think with answering these questions this would be a nice manuscript. It is definitely helpful to contribute to the CHD4-NDD literature, and also supports genome sequencing, or at least sequential testing with exome sequencing.

We appreciate the reviewer’s comments and hope the new version of the manuscript, with all suggestions addressed, is up to par for publication.

Edits:

NDD (neurodevelopmental disorders) is typically used over ND or ND disabilities

We changed all “ND disorder”, “ND disabilities” and “ND condition” to “NDD”: lines 21, 24, 37, 42, 46, 54, 70, 79, 84, 290, 295, 300, 302, 308, 322, 347, 350 and Table 1 (caption and first cell).

Line 28 should read PROC gene

We have changed it accordingly (line 29).

Line 44: I wouldn’t cite a poster for this, and I would list what the most common clinical feature is, as NDDs are typically a large proportion of clinical genetic testing. Or, you can remove this sentence entirely, as it does not change your story.

As suggested, we have removed this sentence from the manuscript.

Line 52: spell out SKDEAS and KBGS. Put OMIM # in parentheses.

We carried out the proposed alteration (lines 54-55). Please note that KBGS stands for KBG syndrome, and that “KBG” are the initials of the family name of the first three affected families that were described. Therefore, this abbreviation does not have a long form.

Line 53: this is a long sentence that reads a little confusing. I would change it to something like “On the contrary, familial BCAs are rarely associated with clinical phenotypes. However, this may not be the case in families with multiple genomic variants or inherited unbalanced translocation variants of BCAs.”

We adapted the suggested changes to the sentence (lines 57-58).

Line 67: again, a long sentence. Maybe change to “Comprehensive genome sequencing (GS) approaches are able to identify the full spectrum of genomic variants, which is valuable towards improved genomic-based healthcare.”  The following sentence is not necessary, of course GS needs GS data analysis/interpt! If you were showing us a novel tool it would make sense to include, but I don’t think its necessary.

We changed the sentences according to the provided suggestions (lines 73-74).

Line 74: use 3.2 Mb instead of 3,195 kb

We have changed it accordingly.

Line 77: include OMIM # (617159)

We have changed it accordingly (line 81).

Line 81: “conventional protocols” is not sufficient, add a citation.

We added the respective citations (line 89, Ref. 7 and 8 of the revised manuscript)

Line 88: saying “essentially” sounds like you’re leaving something out.

We removed the word “essentially” from the sentence, as it was redundant. The protocol was carried as described in the respective reference.

Line 89: is there a github/citation for improperCLAS.py? Otherwise we don’t really need the name of your in house pipeline.

The script improperCLAS.py has a github repository that was added to the manuscript (line 106).

Line 113: remove comma

We have changed it accordingly (line 117).

Case Report: the HPO #s have different formatting (some w/ : some not, some with spaces some not). Be consistent.

All HPO terms were changed to the same formatting (HP:#######) (line 139 and 141, which were incorrectly formatted).

Line 133: “In parallel” doesn’t make sense here. In parallel to what?

The phrase meant “in addition to, at the same time”, but we do agree it is somewhat confusing. We have removed it from the sentence (line 141).

Line 140: “She performed” doesn’t make sense, I would change to “She underwent”. Same with line 144

As suggested, in line 148, “performed” was replaced with “underwent”, while in line 152, it was replaced by “was submitted to”.

Line 164: you say that the mother has “no relevant medical history,” but she is macrocephalic. Is that not relevant, since the proband is too?

 The mother of the proband had only macrocephaly, without any other phenotype or symptom. Clinically, isolated macrocephaly is considered normal variation, which is present in approximately 3% of the healthy general population (this is already imbedded in the definition of macrocephaly, which is a head circumference above the 97th centile, i.e., superior to 97% of the general population) (see “WHO Child Growth Standards: Head circumference-for-age, arm circumference-for-age, triceps skinfold-for-age and subscapular skinfold-for-age, World Health Organization 2007, ISBN 978-92-4-154718-5”). Therefore, isolated macrocephaly is not considered relevant for her medical history.

Line 168: you don’t need to cite GeneCards

We have changed it accordingly (line 177).

Line 169: should read “…is within a low complexity region…”

We have changed it accordingly (line 178).

 Line 170: use kb not bp

We have changed it accordingly (line 180).

Figure 2: It would be good to include the genes encompassed by the duplication in the diagram.

We added the list of OMIM genes encompassed by the duplication to Figure 2. The caption of figure 2 was updated accordingly (lines 222-224).

Below, folded gray arrows indicate the position of OMIM genes in sense and antisense orientations. Their LoF intolerance score, expressed as o/e-ratio of LoF variants is stated below each gene.

Line 224: put # with OMIM # throughout manuscript

We have reviewed all formatting of OMIM numbers throughout the manuscript. The numbers preceded by a “#” symbol refer to a clinical phenotype (lines 54, 55, 84, 265, 266, 274, 275, 293 and Table 1), while those preceded by a “*” symbol refer to a gene (lines 235, 263, 268 and 270-272).

Line 228: https://nmdprediction.shinyapps.io/nmdescpredictor/ suggests that this variant is subjected to nonsense mediated decay, not truncation. Also not sure why you cited the Weiss paper there?

We appreciate the reviewer’s comment. We stated that the variant was predicted to undergo truncation based on the findings of other nonsense/frameshift variants in the DUF 1086 domain, as in the Weiss paper. However, we agree with that using the suggested prediction tool is more sound, and have changed the manuscript accordingly (line 241 and Ref. 13).

Figure 3 is not super clear, maybe align the sequences differently?

Alignment was changed, the figure was simplified and the caption was updated accordingly.

Line 238: italicize GSG1L2. I would also change the sentence to say “GSG1L2, located at the 17p13.1 breakpoint, is predicted to be tolerant to loss-of-function…” Include the pLI score.

We have performed the suggested changes (lines 252-253). The sentence now reads:

The 17p13.1 breakpoint spanning GSG1L2, which predictably encodes an integral component of the plasma membrane, is tolerant to LoF variants Predictably, the 17p13.1 breakpoint spanning GSG1L2 is an integral component of the plasma membrane, tolerant to Loss-of-Function (LoF) variants [o/e = 0.65 (90% CI 0.37 - 1.22); pLI = 0.00]

Genes should be in italics throughout manuscript.

We have revised this throughout the manuscript.

Line 258: spell out AD

Autosomal dominant (AD) is spelled out the first time it appears on the manuscript (lines 53-54).

Line 263: change to “This variant fulfilled the following ACMG SNV criteria… and is considered pathogenic”

We thank the reviewer for the suggestion, and we adapted it to the manuscript (lines 279-285).

Line 274: CHD4-ND should be CHD4-NDD.

We have changed it accordingly, as also suggested in a prior comment.

Line 298: use Mb not bp

We have changed it accordingly (line 315)

Line 304: form should be from

We have changed it accordingly (line 325)

Line 305: but other behavioral features are reported, correct? Perhaps these just haven’t been highlighted in the literature?

Yes, other behavioral features as anxiety or attention deficit hyperactivity disorder were reported in the literature. As explained in the response to the 1st major comment, these features can likely be explained by the variability of the clinical phenotype, as an outcome of the complex neurological phenotype presented by the proband.

We added this idea to the discussion (lines 327-329):

This is likely attributed to the syndrome’s phenotypic variability, to the complex neurological phenotype presented by the proband, but the modulating effect of the large duplication cannot be excluded.

Table 1: cite where the 44 patients are from. You also might want to compare course facies to just dysmorphic features (abnormal facial shape: HP:0001999).

We added the references to the caption of Table 1. As indicated in the Human Phenotype Ontology, Coarse facies is designated as “Absence of fine and sharp appearance of brows, nose, lips, mouth, and chin, usually because of rounded and heavy features or thickened skin with or without thickening of subcutaneous and bony tissues”, while Abnormal facial shape is its superterm that groups together all the facial abnormalities. Therefore, Coarse facies, by definition, is a specific type of facial dysmorphism, which was also reported in OMIM as associated to CHD4-NDD.

I wonder if you could combine Tables 1 and S1? The phenotypic match information is very useful!

We agree with the reviewer’s suggestion. Table 1 and Table S6 (we assumed “Table S1” was a typo) were combined as Table 1.